# Change of Direction Speed and Reactive Agility in Prediction of Injury in Football; Prospective Analysis over One Half-Season

**DOI:** 10.3390/healthcare10030440

**Published:** 2022-02-25

**Authors:** Dragan Mijatovic, Dragan Krivokapic, Sime Versic, Goran Dimitric, Natasa Zenic

**Affiliations:** 1Faculty of Health Sciences, University of Mostar, 88000 Mostar, Bosnia and Herzegovina; mijatovicdragan@ymail.com; 2Faculty for Sport and Physical Education, University of Montenegro, 81400 Niksic, Montenegro; dr.agan@t-com.me; 3Faculty of Kinesiology, University of Split, 21000 Split, Croatia; sime.versic@kifst.eu; 4HNK Hajduk, 21000 Split, Croatia; 5Faculty of Sport and Physical Education, University of Novi Sad, 21000 Novi Sad, Serbia; goran.dimitric@uns.ac.rs

**Keywords:** agility, soccer, flexibility, predictors, outcome

## Abstract

Agility is an important factor in football (soccer), but studies have rarely examined the influences of different agility components on the likelihood of being injured in football. This study aimed to prospectively evaluate the possible influences of sporting factors, i.e., flexibility, reactive agility (RAG), and change of direction speed (CODS), on injury occurrence over one competitive half-season, in professional football players. Participants were 129 football professional players (all males, 24.4 ± 4.7 years), who underwent anthropometrics, flexibility, and RAG and CODS (both evaluated on non-dominant and dominant side) at the beginning of second half-season 2019/20 (predictors). Over the following half-season, occurrence of injury was registered (outcome). To identify the differences between groups based on injury occurrence, *t*-test was used. Univariate and multivariate logistic regressions were calculated to identify the associations between predictors and outcome. Results showed incidence of 1.3 injuries per 1000 h of training/game per player, with higher likelihood for injury occurrence during game than during training (Odds Ratio (OR) = 3.1, 95%CI: 1.63–5.88) Univariate logistic regression showed significant associations between players’ age (OR = 1.65, 95%CI: 1.25–2.22), playing time (OR = 2.01, 95%CI: 1.560–2.58), and RAG (OR = 1.21, 95%CI: 1.09–1.35, and OR = 1.18, 95%CI: 1.04–1.33 for RAG on dominant- and non-dominant side, respectively), and injury occurrence. The multivariate logistic regression model identified higher risk for injury in those players with longer playing times (OR = 1.81, 95%CI: 1.55–2.11), and poorer results for RAG for the non-dominant side (OR = 1.15, 95%CI: 1.02–1.28). To target those players who are more at risk of injury, special attention should be paid to players who are more involved in games, and those who with poorer RAG. Development of RAG on the non-dominant side should be beneficial for reducing the risk of injury in this sport.

## 1. Introduction

From the aspect of kinesiological analysis, football (soccer) is a contact sport of intermittent and poly-structural nature, with high technical-tactical and physical demands [1,2,3]. Each player’s performance is directly influenced by the environment (i.e., teammates, opponents, and ball). To these stimuli, the player must constantly adapt and react in order to assure situational efficiency. Over the last decade, the intensity of the game has increased significantly, and studies have recorded an increase in sprinting distance (speeds above 25 km/h) by approximately 35% over a period of 7 years [1,4]. Additionally, the frequency of games is constantly increasing [5]. As a result of such increases in the psycho-physiological demands, and despite the evident progress of sports medicine, better sports equipment and training grounds and improved recovery methods and prevention programs, injuries are one of the most evident problems in football [6]. Non-contact muscular injuries represent a special problem, which, although being mostly preventable through primary and secondary prevention strategies, account for one third of all time-loss injuries in men’s professional football [7,8,9].

Injuries in professional football represent multiple problems. Firstly, they can remove a player from training and competition for a while, thereby reducing the value of the team and its chances of success. Additionally, they represent a financial burden, either through direct treatment costs or through indirect costs for salaries of the players that are not performing [1,2,10]. In particular, at the elite European level, a one-month absence of a starting player costs the club in question approximately EUR 500,000 on average [11]. Studies have shown that players in the European leagues have an average of two injuries per year and spend around 37 days out of team training [12,13]. All of this has led scientists to study the predictors of injuries in football players to identify potential risks and try to reduce the number of injuries [14].

Recent findings suggest that causes of injuries should be analyzed in a multifactorial and dynamic etiological model with intrinsic (person-related) and extrinsic (environment-related) risk factors, along with training and match workloads [13,14]. Intrinsic factors include the individual biological and psychosocial characteristics of athletes. These, more specifically, include age and gender determinants, muscle strength and flexibility, functional instabilities, imbalances, previous injuries, and maladaptive rehabilitation processes [14].

Many studies have addressed the associations of different motor and functional abilities with injuries [9,14,15]. In general, insufficient muscular strength and endurance, poor cardiorespiratory endurance, and lack of flexibility, power, speed, and balance represent risk factors for musculoskeletal injuries [16,17]. More specifically, in terms of flexibility, higher ranges of motion for the hamstrings and ankles were associated with lower injury risk [9]. There is also moderate evidence that lower levels of body strength, power and balance are good predictors of future muscle injuries [9,15]. Finally, cardiorespiratory endurance showed an association with injury: strong evidence was found that poor performance on timed shuttle runs increased the risk of injury [18]. Interestingly, the authors of a recent comprehensive review, which overviewed the associations between variables of physical fitness and musculoskeletal injury risk, clearly noted that they did not find any study linking agility and injury occurrence in athletes [9].

Specifically, agility represents the ability to quickly and accurately change the direction of motion of the whole body in response to a stimulus [19]. It is generally accepted that agility in professional sports should be graded according to two relatively independent qualities: (i) change of direction speed (CODS) or pre-planned agility, and (ii) reactive agility or non-planned agility (RAG) [20]. From the perspective of injury occurrence, it is important to highlight the specifics of CODS and RAG. While CODS is a change in the speed and direction of movement according to a predetermined pattern, this facet of agility is predominantly influenced by speed, power and anthropometrics [21]. On the other hand, during RAG the athlete must change his direction of movement in response to an external stimulus (i.e., opponent, ball, or teammate), which is logically and additionally influenced by perceptual and cognitive capacities [22,23].

Both CODS and RAG have been repeatedly confirmed as important determinants of success in sports [24,25,26]. This is especially emphasized in team sports, such as football, where the movement of the athlete is directly determined by the movements of teammates, opponents, and the ball itself [27,28,29]. Therefore, there is no doubt that both RAG and CODS capacities are directly responsible for accurate, effective, and (most importantly) safe execution of directional changes, which are themselves known to be movement scenarios where a great deal of injuries occur both in training and in competition [30,31,32]. Surprisingly, there is a lack of studies considering agility as a factor influencing injury occurrence in sports, and to the best of our knowledge no study has reported this association in professional football.

Given the lack of knowledge on the association between agility and musculoskeletal injuries in football, the main aim of this research was to prospectively observe the influences of agility components (RAG and CODS) on injury occurrence in professional football players during one half-season. Additionally, we observed flexibility and sporting factors (playing position, age, experience in football, playing time) as factors of influence on injury occurrence. We hypothesized that RAG and CODS status at the beginning of the season would be inversely related to occurrence of injury, meaning lower risks of injury in players who performed better on football-specific tests of CODS and RAG.

## 2. Materials and Methods

### 2.1. Participants

In this study we observed 129 professional football players from Bosnia and Herzegovina (all males, 24.4 ± 4.7 years of age) All of them were members of teams competing at the highest competitive level in the country, including the national champions for the observed season.

The invitation for study participation was sent by the National Football Federation, and players were informed of potential benefits and risks. Involvement in the research was voluntary, and participants’ personal data were protected with an identification code, known only to the main/head researcher. Prior to the study, participants signed consent forms for participation and the Ethical Board of University of Split, Faculty of Kinesiology approved the investigation (approval number: 2181-205-02-05-14-001). Based on (i) a previously evidenced injury occurrence of 20%, (ii) a population sample of 250 professional players competing in the first national division in Bosnia and Herzegovina (goalkeepers and players younger than 18 years were excluded from total population), (iii) a margin of error of 0.05, and (iv) a confidence level of 0.95, the required sample size for this investigation was calculated to be 125 participants (calculated by Statistica, Tibco Inc., Palo Alto, CA, USA).

Inclusion criteria were: (i) a minimum of 8 years of active football training, (ii) competing at the highest national level, (iii) no injury/illness 15 days prior to baseline testing. Exclusion criteria included: (i) being a goalkeeper, (ii) pain, illness, or injury which prevented the player from performing the tests used in the study, (iii) playing less than 15 min per game prior to occurrence of injury in the following half-season.

This prospective study included baseline testing and during a follow-up period (first competitive half-season in 2019/20) (Figure 1). At baseline (January 2020), we tested predictors, and during the follow-up period (continuously over a period of six months after baseline) we observed outcomes (injury status) (Figure 1).

### 2.2. Variables and Measurements

In this investigation we observed predictors and outcomes. Predictors included age (in years), age when players started to play football (later described as experience in football), anthropometric indices, flexibility, reactive agility, and change of direction speed. Injury status over the competitive half-season was the outcome.

Anthropometrics included (i) body height (in 0.5 cm) (Seca, Birmingham, UK), (ii) body mass (in 0.1 kg) (Tanita TBF-300, Tanita, Tokyo, Japan), both measured with standardized techniques and calibrated equipment, and (iii) calculated body mass index (BMI; in kg/m^2^).

We used two tests of flexibility, the sit and reach test (SIT&REACH), and a test of maximal abduction (MAXABD). The SIT&REACH was commenced under standardized conditions. In brief, the participant was instructed to sit on the floor, with his bare feet placed vertically against a measuring box. He had to lean forward maximally while keeping his knees and arms fully extended, to reach maximally along the measuring tape on the box, and to maintain the final reaching position for approximately 3 s. The measurement was carried out over three trials, and after the reliability analysis showed high consistency of the measurements (intra class correlation (ICC) of 0.96), the highest score was kept as the final result for each participant [33].

The MAXABD was performed using standardized settings and equipment. Each participant sat on the floor, resting his back on the wall, and was instructed to perform maximal voluntary leg abduction while keeping his legs constantly in contact with the floor, and knees fully extended. The maximal distance between the inner malleoli of the right and left legs was measured. The test was performed over three trials and, following the reliability analysis which confirmed appropriate reliability (ICC = 0.93), the maximal score was recorded as the final result for each participant [34].

The CODS and RAG were tested using an originally developed hardware system based on an ATMEL micro-controller (model AT89C51RE2; ATMEL Corp., San Jose, CA, USA). This measurement equipment has been repeatedly used in studies involving athletes from various sports, including football [21,27]. For the measurement, a photoelectric infrared sensor (E18-D80NK) was used as the time triggering input, and LEDs placed in the 30-cm-high cones were used as controlled outputs. Distances used in RAG and CODS were identical (Figure 2), but execution of the tests differed with regard to testing scenarios (e.g., pre-planned and unplanned testing templates—see the following text for details).

For both agility tests, the player ran at maximal intensity through the gate, and when he crossed the infrared signal timing began. At the same moment (no time delay), one cone (either “A” or “B”) was lit. The player had to run at maximum speed in the designated direction, kick the ball with the inside of the foot toward the goal, and then turn and run back as quickly as possible to the starting gate. The timing stopped when the player passed the gate on the way back. For CODS testing, players were informed about which cone would be lit in each trial, and therefore the movement template could be pre-planned. Four trials were performed (A, B, A, B for trials 1–4, respectively), and there was a rest of 20 s between trials. For RAG testing, players did not have information on which cone would be lit up, so their movements were not pre-planned. Despite the fact the players did not know the template of RAG testing in advance, all players were tested using the same sequence throughout the five trials (trial 1: A cone, trial 2: B, trial 3: B, trial 4: A, trial 5: A), with 20 s of rest between trials.

In this study we separately observed performances on dominant and non-dominant sides for both CODS and RAG, as recently suggested [35]. In evidencing dominant and non-dominant side performance, we calculated the average of the first and third trial for CODS (e.g., average performance in the direction of cone A), and the average of the second and fourth trial (average for direction cone B). The better score marked the dominant side, while the poorer result was considered to mark the non-dominant side. Similarly, the average of the first and fourth trial was compared to the average of the second and third trial for RAG, and dominant vs. non-dominant sides were established accordingly. The reliability of the CODS was better than the reliability of the reactive agility (intra class correlations of 0.89 and 0.79 for CODS and reactive agility, respectively), but this was expected and has been explained in detail previously [23].

The Oslo Sports Trauma Research Center Overuse Injury Questionnaire (OSTRC) was used for injury recording [36]. Players were questioned with the OSTRC once a week by the team physician, who sent the results to the main investigator (first author). The outcomes of this study were the incidences of musculoskeletal injuries that occurred during the study in four body regions: ankle, knee, back, and shoulder. Each answer in the OSTRC corresponds to a score. For each question (body location), a score between 0 and 25 is given, and a theoretical score (sum) ranging from 0 to 100 is calculated for four body regions. Reported scores of >39 were classified as injuries. Together with injury reports, physicians reported the total number of hours of training/games for their team. Finally, injuries were reported in total, along with in number of injuries per 1000 h of training/game. Although we observed multiple injuries occurring, for the purposes of statistical analysis and identification of the relationships between predictors and outcome, we did not differentiate single injuries from multiple injuries [37].

### 2.3. Statistics

The Kolmogorov–Smirnov test was used to check for the normality of the distributions for all variables, and all variables but injury occurrence were found to be normally distributed. Consequently, means and standard deviations were reported for all normally distributed variables, while frequencies and percentages were reported for injury occurrence.

*T*-test for independent samples was performed to identify the differences between groups based on injury occurrence. Additionally, differences between injured and non-injured players were evaluated by magnitude-based Cohen’s effect size (ES) statistics (including 95% Confidence Intervals—95%CI) with modified qualitative descriptors, using the following criteria: <0.02 = trivial; 0.2–0.6 = small; >0.6–1.2 = moderate; >1.2–2.0 = large; and >2.0 very large differences.

The associations between the studied predictors and outcomes were evaluated by a logistic regression calculation using binarized criteria on the categorized OSTRC scale (0 = absence of injury, 1 = injury). The odds ratios (OR) with the corresponding 95% CI are reported. In the first phase, predictors were univariately correlated with outcomes. In the next phase, all significant predictors were simultaneously included in the multivariate logistic regression calculation, in order to control the possible confounding effects of the predictors. The Hosmer–Lemeshow test was used to check the model fit (with a significant χ^2^ indicating an inappropriate model fit).

Statistica version 13.5 (Tibco Inc., Palo Alto, CA, USA) was used for all analyses, and a significance level of *p* < 0.05 was applied.

## 3. Results

During the period of this research, 35 players (32%) suffered from injury, while 75 (68%) did not experience any injury as evidenced by the OSTRC questionnaire.

Over approximately 40,000 h of training/game, 51 injuries occurred in total, resulting in 1.3 injury per 1000 h of exposure (training/game). Of all injuries, 61% (32 injuries) occurred during training, while 39% (19 injuries) occurred during games. Since games accounted for 12% of overall players’ exposure during study period, the risk for being injured during a game was three times higher than the risk for being injured during training (OR = 3.1, 95%CI: 1.63–5.88).

A total of 8% of players suffered from multiple injuries. On average, each player suffered from 0.46 injuries over the study period.

Injured players were older (*t*-test = 2.43, *p* < 0.05, moderate ES differences), more experienced in football (*t*-test = 2.02, *p* = 0.02, moderate ES differences), more involved in the game (*t*-test = 4.23, *p* = 0.01, moderate ES differences), and achieved poorer results in RAG-D (*t*-test = 2.03, *p* < 0.05, moderate ES differences) and RAG-ND (*t*-test = 2.47, *p* = 0.02, moderate ES differences) (Table 1).

Results of the univariate logistic regression calculations are presented in Figure 3. Univariate logistic analysis evidenced significant correlations between playing time (OR = 2.01, 95%CI: 1.560–2.58, *p* < 0.001), players’ age (OR = 1.65, 95%CI: 1.25–2.22, *p* < 0.01), experience in football (OR = 1.51, 95%CI: 1.03–2.08, *p* < 0.05), RAG-D (OR = 1.18, 95%CI: 1.04–1.33, *p* < 0.05) and RAG-ND (OR = 1.21, 95%CI: 1.09–1.35, *p* < 0.05), and injury occurrence. Evidently, the risk of injury over the competitive half-season was higher for players who were more involved in active play, older players, those who had more experience in football, and those who performed worse in either reactive agility test.

When all significant predictors were simultaneously included in multivariate logistic regression, significant associations were retained for playing time (OR = 1.81, 95%CI: 1.55–2.11, *p* < 0.001) and RAG-ND (OR = 1.15, 95%CI: 1.02–1.28, *p* < 0.05), with proper model fit as indicated by Hosmer Lemeshow test (*p* > 0.05). Specifically, players who were more involved in the game and who performed poorly in the reactive agility task on the non-dominant side were more prone to injury over the study period (Figure 4).

## 4. Discussion

The main aim of this study was to investigate the influences of RAG and CODS agility components on injury occurrence in professional football players, and we hypothesized lower risks of injury in players who performed better on the football-specific tests of CODS and RAG. Regarding this, there are several important findings of the study. First, at the univariate level, older age, longer playing time, and poor performances in both RAG tests (RAG-ND, and RAG-D) were identified as risk factors for injury occurrence. However, the multivariate model identified longer playing time and poorer RAG-ND as the only significant risk factors for injury occurrence. As a result, our initial hypothesis may be partially accepted.

### 4.1. Age and Playing Time as Predictors of Injury

Our results suggest that older football players are more prone to injuries, and this is generally supported by previous reports. Even studies carried out 30 years ago have shown that the incidence of injuries in football increases with the age of the players (0.0331 versus 4.021 injuries in youth and professional players, respectively) [38]. This difference can be seen even in younger age categories, where the incidence of injuries is five times lower among athletes aged 7 to 12 compared to those older than 12 [38]. This is not characteristic only for males, as age has been shown to be a significant risk factor in a prospective study of female football players also [39]. Although it is clear that injury incidence will be affected by the large difference in intensity between training and games (through it increases in both settings with age), older players will undoubtedly be exposed to higher training loads over the years, which itself increases the risk of injury [38].

A prospective cohort study of 23 elite European teams showed that over a 7-year period, a significantly higher incidence of injuries was observed in matches compared to training sessions (27.5 vs. 4.1, per 1000 h of exposure) [12]. Supportively, results of our study suggest that active involvement in the game (e.g., playing time) increases the risk of injury. However, it is important to note that in our multivariate logistic regression model (when all significant predictors were simultaneously included in the model), playing time actually diminished the influence of age on injury occurrence. In other words, older (i.e., more experienced) players spend more time playing in games and play more matches, which naturally increases their likelihood of being injured [12,40,41,42].

It is relatively well documented that that a larger amount of exposure to soccer directly correlates with the number of injuries [43,44]. A study of 41 teams in the four best divisions in Sweden stated that exposure to soccer and ankle sprain injuries per player were more frequent in higher divisions [44]. Similar findings were presented in a study of female soccer players, where multivariate logistic regression showed that higher exposure to soccer significantly increases the risk of traumatic leg injuries [43]. Additionally, a study of South African Premier Soccer League clubs showed that, besides other factors, exposure time to soccer has an impact of the risks and mechanisms of injuries [45]. Therefore, our results are generally supported by previous considerations regarding playing time as a risk factor for injuries in soccer.

### 4.2. Reactive Agility and Injury Occurence

One of the most important findings of this study was that players who achieved better results on RAG tests had lower incidences of injury. At the same time, results on the CODS tests were not significantly related to the frequency of injuries. In analyzing these findings, the differences between determinants of CODS and determinants of RAG should be considered. In brief, CODS is primarily influenced by motor capacities (e.g., speed, power, and coordination) and anthropometrics. On the other hand, the importance of these factors is not so pronounced in RAG, where perceptual and decision-making skills (e.g., visual scanning, anticipation, pattern recognition and knowledge of situations) are highly important [19]. It is therefore reasonable to conclude that players who achieved better results on RAG generally have better perception of the environment during games and react better to new situations.

The movements of players during a football game are constantly influenced by the movement of teammates, opposing players and the ball. As a result, faster and better decision making is expected to be a protective factor against dangerous and risky situations. More specifically, a player with better RAG will be able to recognize a potentially risky situation beforehand, and to avoid unnecessary duels or collisions with the opponent. Additionally, a player who has better RAG can react in a timely manner and prevent additional high-intensity stress to his muscles, such as sprinting, sudden breaking, and similar movements that risk injuries [7].

A recent systematic review of the associations between physical fitness and muscle injuries has found that certain levels of flexibility, power, speed, and balance present risk factors for injury occurrence, but the authors stated that no study has analyzed the relations between agility and muscle injuries in football [9]. That said, we did find one study where the authors examined the influence of agility on injury occurrence in Australian rugby. In this study, rugby players with poorer RAG performances, specifically longer decision times, had a lower risk of injury [46]. These results were surprising, even for the authors themselves, who originally expected that RAG would be protective against injuries. However, their explanation was that players with poorer reaction times can inadvertently avoid severe collisions which lead to injuries and have just partial contact with opponents which does not result in full force tackles [46].

Although the finding from the previously cited Australian rugby study disagrees with our findings (we found a lower injury risk for players with better RAG), several important facts could explain the disagreement. First of all, rugby is a contact sport where frequent hitting and tackling occurs, and in the aforementioned study, contact injuries were analyzed [46]. On the other hand, in this research, both contact- and non-contact injuries were analyzed. Furthermore, in the study on rugby players, the playing time was not analyzed as a predictor of injuries. Therefore, it was possible that those players who spent more time in the game had better RAG, and because of that were simply more exposed to high-risk situations. This is particularly important if we consider that better RAG in rugby is closely related to the qualitative level of players, and therefore it can be assumed that these players will spend more time in game play [47,48]. Finally, in the rugby study, a significant predictor of injuries was “decision time during RAG” (and not RAG itself). Although an important determinant of RAG, “decision time” is actually only one part of RAG performance. On the other hand, herein we used a sport-specific RAG test that includes a specific ball manipulation task, which entails a much more complex movement pattern, all the more so considering the dominant and non-dominant-side testing.

### 4.3. Dominant vs. Non-Dominant Side Reactive Agility Performances and Injury Occurence

RAG-ND stands out as the more significant predictor of injury occurrence than RAG-D. More specifically, when we calculated multiple logistic results, the RAGND was a significant predictor of injury occurrence, together with playing time. The first reason for the fact that RAGD was not a significant predictor in the multivariate calculation can probably be found in the level of the players (professional players with more than 10 years of experience in systematic football training). Therefore, all participants were highly skilled, and their dominant-side performances (irrespective of the unplanned nature of the task for RAG) were highly efficient. On the other hand, execution of the RAG test on the non-dominant side actually presents the “real-world” unpredictable movement template and is logically more related to eventual discrepancies in reactive agility tasks on the field [49].

Specifically, RAG represents coping in emerging situations [50]. Since the RAG-D test for our players enabled them to use their highly refined skills, the RAG-ND was more likely to be a significant predictor of injury, since it was more likely to be representative of unexpected circumstances. Indeed, during the game, footballers have 1200–1400 changes of direction, which occur on average every 2 to 4 s [51]. These events include path changes, jumping, acceleration, and deceleration, and themselves represent risky movements in terms of injury [52,53]. Since most of these high-risk-movements take place under the influence of external stimuli (i.e., reactively), the importance of RAG (especially when executed on the non-dominant side) in the prevention of injuries is clearly substantial.

### 4.4. Limitations and Strengths

The most important limitation comes from the fact that we observed players over one half-season only. Therefore, we lacked data on injuries later in the season. Second, in this study we used OSTRC as an overall index of injury status. There is no doubt that some of the injuries recorded by the OSTRC in our study could not be (even theoretically) connected to the studied motor capacities but are clearly related to collisions and contact (shoulder injuries, for example). However, we were of the opinion that the arbitrary elimination of some injuries from the record would have resulted in even greater error, so we included all recorded injuries. Additionally, our study involved professional football players from only one country, and during a specific period of time (a season which was interrupted by COVID-19 lockdown). Due to the great deal of contextual factors which are known to be related to specifics of the sport (even injury occurrence), the results are generalizable to samples similar to the one observed in our research.

This is one of the first studies where different agility components were related to injury occurrence in professional team sports, and to the best of our knowledge the first one where this was done for football. We observed professional football players competing at the highest level of national competition. The usage of the football-specific tests of CODS and RAG, and separate evaluations of the performances on the dominant and non-dominant sides, are important strengths of the study. Therefore, although this investigation is not the final word on a topic, we hope it will increase knowledge and initiate further studies.

## 5. Conclusions

This study highlighted the importance of RAG as a predictor of injury occurrence. Generally, professional players perform numerous changes in direction per game in pre-planned and unplanned scenarios, and these manoeuvres are often performed rapidly when they do not have full control over their bodies. Therefore, it is clear that scenarios requiring reactive agility present particularly risky conditions for dysfunctional locomotion, and consequently may result in injury. As a result, coaches working with football athletes should pay particular attention to mastering technical movement skills which will assure musculoskeletal safety in RAG tasks.

Special attention should be paid to RAG performance on the non-dominant side. Specifically, we studied professional players, and almost certainly their dominant-side RAG performances were highly refined. In other words, despite the differences in RAG-D performance, the technical quality of the RAG-D execution for most of the players was almost certainly proficient, and therefore safe. Meanwhile, RAG-ND was found to be an important determinant of injury occurrence, even when statistically controlled for the most significant factor of influence—playing time. This clearly points to the necessity of: (i) independent evaluation of RAG performances on dominant and non-dominant sides, and (ii) specific mastering of RAG on the non-dominant side in professional football players.

Playing time was found to be most important risk factor for injury occurrence in the studied players. While games present the most important psycho-physiological stress on players, resulting in large numbers of injury-risk situations, such results are expected and in line with previous reports. As a result, in order to decrease the risk of injury in professional football, special attention should be paid to players who are on the field during games for a long time, especially if they are older and lack RAG.

## Figures and Tables

**Figure 1 healthcare-10-00440-f001:**
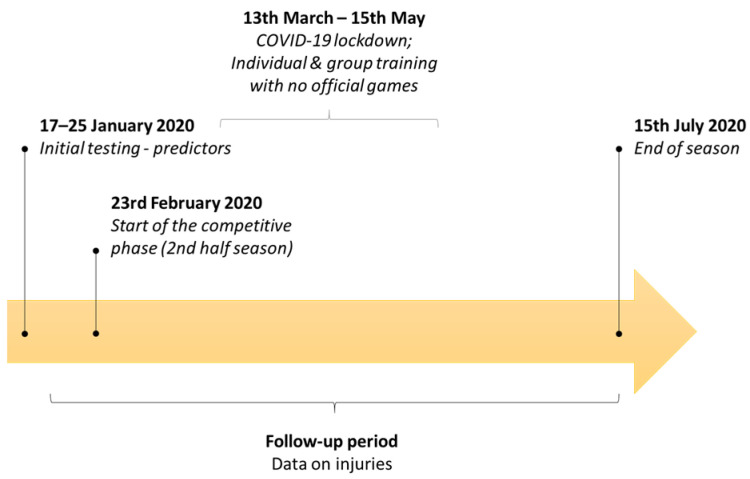
Study design.

**Figure 2 healthcare-10-00440-f002:**
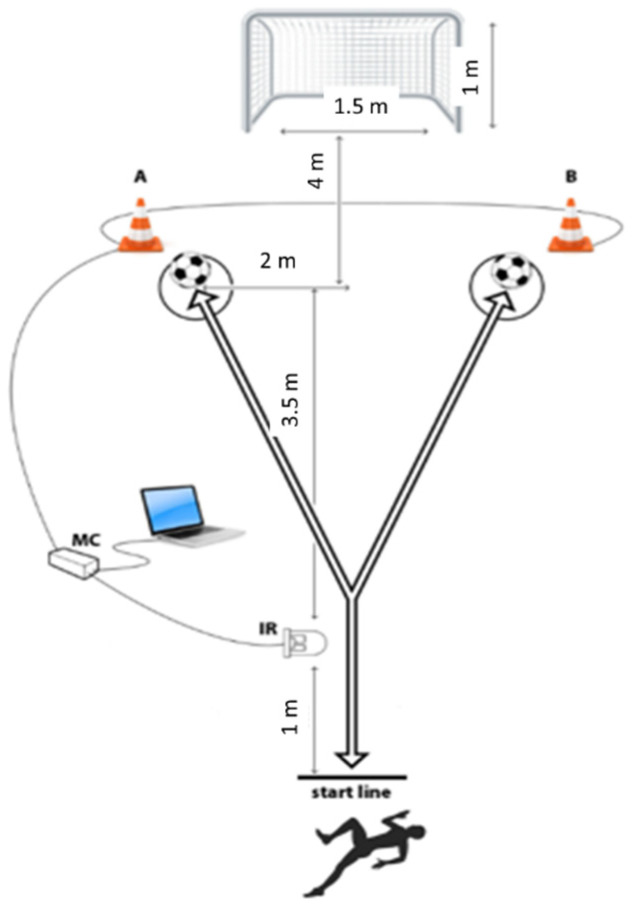
Testing of reactive agility and change of direction speed in football.

**Figure 3 healthcare-10-00440-f003:**
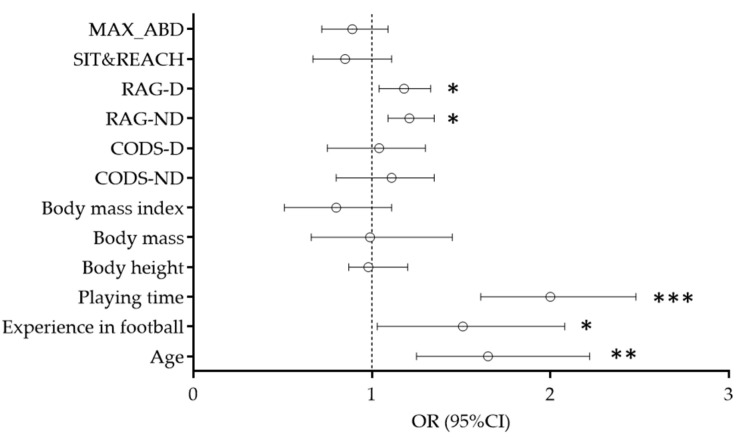
Results of the univariate logistic analysis for prediction of injury occurrence (*** *p* < 0.001, ** *p* < 0.01, * *p* < 0.05).

**Figure 4 healthcare-10-00440-f004:**
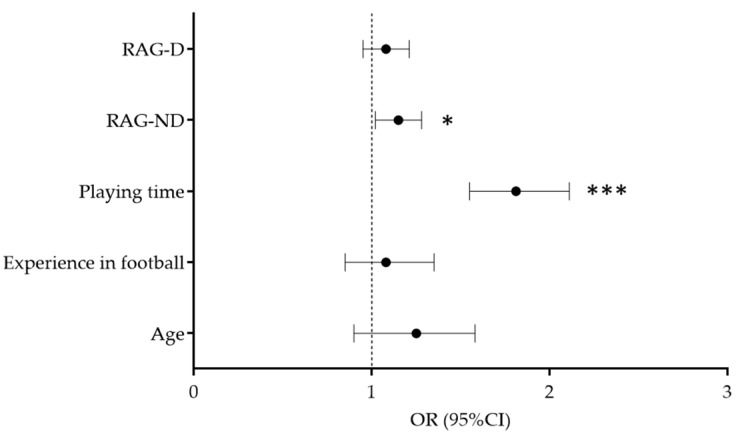
Results of the multivariate logistic analysis in prediction of injury occurrence (*** *p* < 0.001, * *p* < 0.05).

**Table 1 healthcare-10-00440-t001:** Descriptive statistics and differences in study variables between injured and non-insured players.

Variables	Injured (*n* = 35)		Non-Injured (*n* = 75)		*t*-test		Effect Size
	Mean	SD	Mean	SD	*t*-test	*p*	d (95%CI)
Age (years)	26.2	4.4	23.8	4.8	2.43	0.02	0.51 (0.11–0.92)
Experience in football (years)	16.97	2.93	14.79	2.87	2.02	0.03	0.75 (0.34–1.16)
Playing time (min/game)	67.11	7.03	58.98	8.98	4.23	0.01	0.96 (0.54–1.38)
Body height (cm)	187.87	6.78	185.25	7.21	0.25	0.79	0.37 (−0.03–0.77)
Body mass (kg)	78.99	7.51	79.00	6.25	0.01	0.99	0.01 (−0.4–0.40)
Body mass index (kg/m^2^)	23.32	1.23	23.26	1.18	0.22	0.81	0.05 (−0.35–0.45)
CODS-ND (s)	2.87	0.25	2.89	0.31	0.24	0.78	0.06 (−0.33–0.46)
CODS-D (s)	2.74	0.2	2.76	0.39	0.3	0.74	0.06 (−0.34–0.46)
RAG-ND (s)	3.15	0.20	2.94	0.24	2.47	0.02	0.92 (0.50–1.34)
RAG-D (s)	3.08	0.19	2.94	0.25	2.03	0.04	0.66 (0.25–1.07)
SIT&REACH (cm)	30.26	7.51	29.69	7.39	0.34	0.72	0.07 (−0.32–0.47)
MAXABD (cm)	138.35	12.79	138.57	13.25	0.07	0.8	0.016 (−0.38–0.42)

Legend: CODS—change of direction speed, RAG—reactive agility, D—dominant side, ND—nondominant side, SIT&REACH—sit and reach flexibility test, MAXABD—maximal abduction flexibility test

## Data Availability

Data will be provided to all interested parties upon reasonable request.

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
