# Peer review of "Change of Direction Speed and Reactive Agility in Prediction of Injury in Football; Prospective Analysis over One Half-Season"

_healthcare, 2022, doi:10.3390/healthcare10030440_

Round 1
Reviewer 1 Report
The manuscript is clearly written. The manuscript is acceptable in its present form because seems sufficiently promising to be published. However, two suggestions may be considered:
Line 40: please remove “aggression”, because is not the adequate term to express that idea.
Line 115-116: There is no signed consent of participants? Regarding special ethical concerns arising from the use of human subjects, this is a key aspect.
1. The aim of the study is addressed by the research design and results. 2. Topic is relevant as injuries in soccer are a hot topic in literature. Moreover, results may improve training prescription with this population.3. New insights are open with its publications, as results obtained add factors that need to be considered in injury prevention and rehabilitation.
4. This manuscript is methodologically correct.
5. The aim of the study is addressed by the research design and results.
6. References are adequately redacted and include some of the most important studies about this topic.
7. Figures and tables are clear, and are complementary to the main text information.
Author Response
The manuscript is clearly written. The manuscript is acceptable in its present form because seems sufficiently promising to be published. However, two suggestions may be considered:
We would sincerely like to thank You for your review and recognizing the value of our study. We amended the manuscript in accordance with your suggestions.
Line 40: please remove “aggression”, because is not the adequate term to express that idea.
Response: Thank You, the word “aggression” is moved from the text.
Line 115-116: There is no signed consent of participants? Regarding special ethical concerns arising from the use of human subjects, this is a key aspect.
Response: Thank You for noticing. The participants, of course, signed consent form for participation in the research and this information has been added to the text (please see participants section in methods chapter).
- The aim of the study is addressed by the research design and results. 2. Topic is relevant as injuries in soccer are a hot topic in literature. Moreover, results may improve training prescription with this population.
- New insights are open with its publications, as results obtained add factors that need to be considered in injury prevention and rehabilitation.
- This manuscript is methodologically correct.
- The aim of the study is addressed by the research design and results.
- References are adequately redacted and include some of the most important studies about this topic.
- Figures and tables are clear, and are complementary to the main text information.
Thank you for your suggestions and comments.
Staying at your disposal
Reviewer 2 Report
Change of direction speed and reactive agility in prediction of injury in football; prospective analysis over one half-season
First of all, the reviewer would like to thank the authors for their work and efforts in trying to improve sports science knowledge.
General comments to the authors
Overall, this is a nice study that could have great prospective analysis over one half-season when integrated with soccer players evaluating change of direction speed and reactive agility in prediction of injury in football. The authors are commended on their efforts thus far. The study is well designed and well-written, with a great original article evaluating the usefulness of the topic. However, I suggest only small corrections and the authors should update the recent references about the soccer, these corrections and studies will allow improving the manuscript.
Abstract
This section is well designed and well-written.
Introduction section
Page 1 line 39: instead of he, you should use the players or they
Methods section
You should add the ethics number
You should add the references for SIT&REACH and MAXABD tests.
You should add the description of effect size in statistical analyses and add effect size in table 1.
Results section
This section is well designed and well-written.
Discussion section
Overall the discussion is well-written and incorporates relevant literature.
Figures and Tables
This section is well designed and well-shown.

Author Response
First of all, the reviewer would like to thank the authors for their work and efforts in trying to improve sports science knowledge.
General comments to the authors
Overall, this is a nice study that could have great prospective analysis over one half-season when integrated with soccer players evaluating change of direction speed and reactive agility in prediction of injury in football. The authors are commended on their efforts thus far. The study is well designed and well-written, with a great original article evaluating the usefulness of the topic. However, I suggest only small corrections and the authors should update the recent references about the soccer, these corrections and studies will allow improving the manuscript.
Thank You very much for your review. We highly appreciate your suggestions and we revised the manuscript accordingly.
Abstract
This section is well designed and well-written.
Response: Thank You.
Introduction section
Page 1 line 39: instead of he, you should use the players or they
Response: Thank You, the text is amended accordingly.
Methods section
You should add the ethics number
Response: Thank You for noticing, the ethical approval number is added. Text reads: “Prior to study, participants signed consent forms for participation and the Ethical Board of University of Split, Faculty of Kinesiology approved the investigation (approval number: 2181-205-02-05-14-001). “ (please see participants section in methods chapter).
You should add the references for SIT&REACH and MAXABD tests.
Response: Thank You, the references are added. Specifically
- López-Miñarro, P.A.; de Baranda Andújar, P.S.; RodrÑGuez-GarcÑa, P.L. A comparison of the sit-and-reach test and the back-saver sit-and-reach test in university students. Journal of sports science & medicine 2009, 8, 116.
- Malliaras, P.; Hogan, A.; Nawrocki, A.; Crossley, K.; Schache, A. Hip flexibility and strength measures: reliability and association with athletic groin pain. British journal of sports medicine 2009, 43, 739-744.
You should add the description of effect size in statistical analyses and add effect size in table 1.
Response: As suggested in this version of the manuscript we included effect size statistics, and it is now presented in last column in the Table 2 (please see collumn highlighted in yellow), and Results section. Specifically, text reads: “Injured players were older (t-test = 2.43, p < 0.05, moderate ES differences), were more experienced in football (t-test = 2.02, p = 0.02, moderate ES differences), they were more involved in game (t-test = 4.23, p = 0.01, moderate ES differences), achieved poorer results in RAG-D (t-test = 2.03, p < 0.05, moderate ES differences), and RAG-ND (t-test = 2.47, p = 0.02, moderate ES differences) (Table 1).” (please see 4th paragraph of the Results section).
Results section
This section is well designed and well-written.
Response: Thank You.
Discussion section
Overall the discussion is well-written and incorporates relevant literature.
Response: Thank You very much for recognizing the values of our main findings and their application in practice,
Figures and Tables
This section is well designed and well-shown.
Thank you for your suggestions and comments.
Staying at your disposal
Reviewer 3 Report
Review for HC
Change of direction speed and reactive agility in prediction of injury in football; prospective analysis over one half-season
General comments
The topic taken up by the authors is important from the point of view of the development of areas like a sports science and from the point of view of sports practice. The authors used reliable tests and research tools. The manuscript uses appropriate, important citations from the scientific literature. After some minor corrections, I think the article should be published
Specific comments
Abstract
Page 1 Line 18-21, how many competitors were participated in this study. It does not follow from this text, please add this information and correct the sentence because it is not linguistically clear
Page 1 Line 22, there is no information in the abstract on what statistical analysis was used in this research project. Please add
Introduction
the introduction is well structured with a good justification by the authors
Methods
Although the description of the tests somewhat explains how the attempts were made, I strongly recommend improving the readability and accuracy of Figure 2. For example, adding a player, adding gates, what was the likely line of movement of the player, goal height ect.
Results
I also see a lot of room for improvement in Table 1 and Figures 3 and 4. Please mark the significance
Discussion
The discussion is generally well written at least in section 4.3. Dominant vs. non-dominant side reactive agility performances and injury occurrence, two more references to literature should be added
A paragraph about limitations is ok
Conclusions drawn on the basis of the obtained results
The manuscript uses appropriate, important citations from the scientific literature
Author Response
General comments
The topic taken up by the authors is important from the point of view of the development of areas like a sports science and from the point of view of sports practice. The authors used reliable tests and research tools. The manuscript uses appropriate, important citations from the scientific literature. After some minor corrections, I think the article should be published
We are very glad that You find our research as value addition for sport science and practice. Thank You for all suggestions and comments, we followed them and we believe that we improved our manuscript.
Specific comments
Abstract
Page 1 Line 18-21, how many competitors were participated in this study. It does not follow from this text, please add this information and correct the sentence because it is not linguistically clear.
Response: Thank You for your comment, we amended the text and it now reads: “Participants were 129 football professional players (all males, 24.4±4.7 years), underwent anthropometrics, flexibility, and RAG and CODS (both evaluated on non-dominant and dominant side) measurements at the beginning of second half-season 2019/20.” (please see text highlighted in yellow).
Page 1 Line 22, there is no information in the abstract on what statistical analysis was used in this research project. Please add
Response: Thank You, the text is added and reads: “To identify the differences between groups based on injury occurrence T-test was used and logistic regression was calculated for associations between the studied predictors and outcomes.”
Introduction
the introduction is well structured with a good justification by the authors
Response: Thank You for Your comment.
Methods
Although the description of the tests somewhat explains how the attempts were made, I strongly recommend improving the readability and accuracy of Figure 2. For example, adding a player, adding gates, what was the likely line of movement of the player, goal height ect.
Response: Thank You. We amended the figure and added a player and goal height and we believe the figure is now improved and easy to understand for the readers.
Results
I also see a lot of room for improvement in Table 1 and Figures 3 and 4. Please mark the significance
Response: Thank you for your comment. The significance of the ORs is indicated in the Figures 2 and 3, and explained in the Legend of the Figures.
Discussion
The discussion is generally well written at least in section 4.3. Dominant vs. non-dominant side reactive agility performances and injury occurrence, two more references to literature should be added
Response: Thank You, the references are added:
- McNeil, D.G.; Spittle, M.; Mesagno, C. Imagery training for reactive agility: Performance improvements for decision time but not overall reactive agility. International Journal of Sport and Exercise Psychology 2021, 19, 429-445.
- Jeffreys, I. A task-based approach to developing context-specific agility. Strength & Conditioning Journal 2011, 33, 52-59.
A paragraph about limitations is ok
Response: Thank You
Conclusions drawn on the basis of the obtained results
Response: Thank You for the recognition of our major conclusion points.
The manuscript uses appropriate, important citations from the scientific literature
Response: Thank You for your comment.
Thank you for your suggestions and comments.
Staying at your disposal!